# Effect of Different Enological Tannins on Oxygen Consumption, Phenolic Compounds, Color and Astringency Evolution of Aglianico Wine

**DOI:** 10.3390/molecules25204607

**Published:** 2020-10-10

**Authors:** Luigi Picariello, Alessandra Rinaldi, Martino Forino, Francesco Errichiello, Luigi Moio, Angelita Gambuti

**Affiliations:** 1Department of Agricultural Sciences, Grape and Wine Science Division, University of Naples “Federico II”, Viale Italia (Angolo Via Perrottelli), 83100 Avellino, Italy; luigi.picariello@unina.it (L.P.); alessandra.rinaldi@unina.it (A.R.); forino@unina.it (M.F.); francescoerrichiellojr@gmail.com (F.E.); luigi.moio@unina.it (L.M.); 2Biolaffort, 126 Quai de la Souys, 33100 Bordeaux, France

**Keywords:** enological tannins, tea tannins, red wine, astringency, oxidation, astringency sub-qualities

## Abstract

Background: In the wine industry, in addition to condensed tannins of grape origin, other commercial tannins are commonly used. However, the influence of oxygen uptake related to different tannin additions during the post fermentative phase in wine has not been completely investigated. In this study, we evaluated the influence of four different commercial tannins (namely, condensed tannins, gallotannins, ellagitannins and tea tannins) during four saturation cycles. Method: Wine samples were added with four different tannin classes (30 g/hL) as to have 5 different experimental samples: control, gallotannins (GT), condensed tannins (CT), ellagitannins (ET), and tea tannins (TT). The chemical composition of the four commercially available tannin mixtures was defined by means of NMR and high-resolution mass spectrometry. After the addition of tannins, each wine sample was oxidized by air over four cycles of saturation. During the experiment oxygen consumption rate (OCR), sulfur dioxide consumption, acetaldehyde production, phenolic compounds, chromatic characteristics, astringency measured by the reactivity towards saliva proteins and astringency subqualities were evaluated. Results: The experiment lasted 52 days. The addition of tannins influenced the oxygen consumption on the 1st day of the saturation cycles and, in the case of TT, a higher total consumption of oxygen was also detected. Acetaldehyde increased during the experiment while the native anthocyanins decreased throughout the oxidation process. Conclusion: Wines added with tannins featured improved color intensities with respect to the control; the addition of TT, GT and ET slightly promoted the formation of short polymeric pigments; the astringency, determined before and at the end of the experiment, decreased in all the samples, including the control wine, and mostly in the ET and GT samples.

## 1. Introduction

The use of enological tannins is a common practice in winemaking, authorized by the International Organization of Vine and Wine (OIV) to facilitate the clarification of wines and musts. They are also used for many other purposes, such as inhibiting laccase in Botrytis-infected grapes [1], reducing the use of sulfur dioxide [2], and stabilizing the color of red wines [3].

In the wine industry, in addition to condensed tannins of grape origin, other commercial tannins are commonly used [4], including hydrolyzable tannins, gallotannins and ellagitannins, which are all extracted from different botanical species [5]. Ellagitannins are obtained from oak and chestnut wood [6], while gallotannins are mainly from galla and tara nuts [7]. Tea tannins, well known for their beneficial bioactivities [8], were also tested in model wine solutions by Ugliano, Slaghenaufi, Picariello and Olivieri 2020 [9], who showed that they were capable of rapidly consuming oxygen, although this is associated with an increased decline in the SO_2_ content.

Recently, some studies have reported on the importance of the use of enological tannins during oxidative processes. In model solutions, it was observed that gallotannins have an excellent scavenging capacity while the ellagic improve iron(II) chelation and the condensed tannins have a significant ability to scavenge peroxyl radicals [10]. Moreover, the addition of ellagitannin in red winemaking operations increased the rate of O_2_ consumption [11]. The consumption of oxygen in wine is strictly related to phenolic compounds which are primary substrates for wine oxidation [12]. Wine oxidation occurs thanks to the presence in wine of transition metal ions and reducing species with a cathecol-like structure (*o*-diphenol groups). When oxidation reactions start, the quinone is generated, and oxygen is reduced to hydrogen peroxide. Quinones and hydrogen peroxide are the main players of further processes of oxidation. In particular, hydrogen peroxide reacts with Fe^2+^ via the Fenton reaction to produce highly reactive hydroxyl radicals, which, in turn, oxidize additional wine constituents, including the most abundant, the ethanol to produce acetaldehyde. Quinones react with several wine nucleophiles including thiols and the tannin phloroglucinol group [13]. Sulfur dioxide can react with quinones and acts as a quencher for the hydrogen peroxide stopping the oxidation process. On the other hand, when acetaldehyde is produced, several reactions with phenolics and other nucleophilic species occurred [14]. In red wines, most of them involve anthocyanins and flavanols and can give ethyl linked phenolic compounds, stabilizing the red color and, often, changing the mouthfeel properties of wine tannins [15]. Obviously, differences in specific wine phenolic composition account for great differences among oxygen consumption rate and the effect on wine quality, as observed when wines with different anthocyanins/tannins ratio underwent a controlled oxygen exposure. [16,17].

In regard to enological tannins, Ugliano, Slaghenaufi, Picariello and Olivieri [9,18] ascertained that different types of commercial tannins increase the oxygen consumption rate (OCR) and affect sulfur dioxide during an oxidative process in model wine solution. Moreover, Rinaldi and Moio [19] showed that the addition of different types of enological tannins affects the astringency and determines different astringency subqualities in Sangiovese wine. However, no information on the evolution of treated wines has been shown yet.

Therefore, evaluating the activity of enological tannins in a more complex mixture such as real wine would allow one to define the actual action that these compounds exert in wine, thus advancing the knowledge about the ability of enological tannins to influence the production of major oxidation products, such as acetaldehyde.

The aim of this work was to evaluate the influence of four different commercial tannins (condensed tannins, gallotannins, ellagitannins and tea tannins), during four saturation cycles, on the oxygen consumption rate (OCR), sulfur dioxide consumption, acetaldehyde production, phenolic compound contents and color evolution in Aglianico wine. Additionally, astringency was measured by the reactivity of tannins towards salivary proteins (Saliva Precipitation Index (SPI)) and astringency subqualities evaluated in the employed red wine subjected to oxidation.

The applications of this study, however, could be important to understand the implications and interactions with different parameters above reported in real wine and to evaluate the real applications of different oenological tannins in wine industry.

## 2. Results and Discussion

In this study, the effect of enological tannins on the oxygen consumption rate was evaluated in real winery conditions with the purpose of investigating the evolution of wine during aging. Because reactions occurring in wines strongly depend on tannin composition, a chemical characterization of the enological tannins used during the study was performed.

### 2.1. Chemical Characterization of Enological Tannins Used

Four commercially available tannin mixtures, referred to as ET, TT, GT and CT throughout the text, were analyzed (pure tannins) by ^1^H-NMR and high-resolution mass spectrometry (HR-MS). ^1^H-NMR spectra provided key insights into the chemical structure of molecules occurring in the mixtures either as monomers or subunits of larger oligomers (Appendix A). The parallel interpretation of HR-MS spectra acquired in the negative ion mode with a mass tolerance of 10 ppm led us to achieve a double goal. First, the NMR-based identification of the monomers present in each tannin mixture was confirmed, and, secondly, the degree of polymerization of each tannin class was assessed.

The parallel NMR- and MS-based analysis conducted on each tannin mixture allowed us to ascertain that both CT (Appendix A) and TT (Appendix A) mixtures contained tannins constituted by condensed flavanols with different composition and polymerization degree, while GT (Appendix A) and ET (Appendix A) turned out to be hydrolyzable tannins (Table 1).

The ^1^H-NMR spectrum of the ET mixture (Appendix A) contained typical resonances of galloyl subunits along with characteristic carbohydrate signals, mainly attributable to glucose [20]. Thus, the molecules contained in the ET mixture were unambiguously identified as ellagitannins. The identity of several of them was tentatively proposed as reported in Table 1, on the basis of HR-MS analysis (Appendix A).

Likewise, the ^1^H-NMR spectrum of the TT mixture (Appendix A) showed resonances assigned to epigallocatechin-3-*O*-gallate (EGCG), along with another set of less intense proton signals attributable to epigallocatechin (EGC) [21]. The relative abundance of EGCG and EGC was assessed by a comparative integration of resonances relative to H3 of either flavan-3-ol (5.50 ppm for EGCG and 4.19 ppm for EGC), since such signals appeared sufficiently isolated for accurate integration. The amount of EGCG was approximately twice that of EGC. Other potential flavan-3-ols did not occur in high enough amounts to be unambiguously identified by NMR. The analysis of the HR-ESIMS spectrum of a sample of the TT mixture (Appendix A) allowed us to confirm the occurrence of EGCG (*m/z* 457.0742 corresponding to C_22_H_17_O_11_) and of EGC (*m/z* 305.0645 corresponding to C_15_H_13_O_6_). Moreover, investigation of the ion peaks contained in the mass spectrum brought to light the occurrence of a number of molecule species in the mixture derived from the polymerization of both EGCG and EGC, as displayed in Table 1. It is necessary to underline that the molecular weight of EGCG and EGC is not different from that of gallocatechin-3-*O*-gallate (GCG) and gallocatechin (GC), respectively; thus, theoretically, EGCG is interchangeable with GCG and, likewise, EGC is interchangeable with GC. Nonetheless, given that neither GCG nor GC were clearly detected in the NMR spectrum, we referred to the identified molecules to be solely constituted by EGCG and EGC, even if a possible substitution of either flavan-3-ol with the corresponding diastereoisomers, GCG and GC, cannot be ruled out.

In the ^1^H-NMR of the GT mixture, we identified typical resonances attributable to quinic acid and to gallate subunits [22]. Thus, we inferred the occurrence of polygalloyl quinic acid esters (Appendix A). Once again, as discussed above, the analysis of the HR-MS spectrum acquired in the negative ion mode (Appendix A) was crucial in determining the number of galloyl subunits bound to a central quinic acid moiety (Table 1).

The ^1^H-NMR spectrum of the CT mixture (Appendix A) showed broad and not well resolved resonances indicative of the presence of high molecular weight polymers as opposed to the TT spectrum where monomers, namely EGCG and EGC, appeared to be by far the most abundant constituents. By comparison with NMR data reported for catechin (C) and epicatechin (EC) [23], in the mixture spectrum we could identify signals attributable to either flavan-3-ol approximately in a 1:1 ratio. Additionally, resonances typical of carbohydrates including glucose and fructose were also detected. As discussed for the TT mixture, the analysis of the HR-MS spectrum acquired in the negative ion mode (Appendix A) allowed to confirm the presence of both C and EC, sharing the same molecular weight (*m/z* 289.0697) as well as to reveal the occurrence of (E)GC (*m/z* 305.0644) and of (epi)catechin-3-*O*-gallate ((E)CG) (*m/z* 441.0794). Additionally, several condensed tannins were identified as displayed in Table 1. By calculating the number of unsaturations (RDB) for each molecular formula, suggested by the *m/z* value with a mass tolerance of 10 ppm, we could also determine the number of potential A-type bonds occurring in the identified polymers. As shown in Table 1, we detected prominent ion clusters in the spectrum regions ranging from 570 to 650 Da corresponding to dimers, from 860 to 890 Da corresponding to trimers, from 1145 to 1170 Da corresponding to tetramers, from 1430 to 1460 corresponding to pentamers, and eventually from 1720 to 1750 Da corresponding to hexamers. The occurrence of larger polymers than those reported in Table 1 cannot be ruled out, as the employed mass spectrometer was set to detect ions peaks up to *m/z* 2000 and no multicharged ions, possibly indicative of polymeric molecules larger than 2000 Da, were present in the recorded mass spectra.

### 2.2. Oxygen Saturations Kinetics and Sulfur Dioxide Consumption

Wines containing the different types of tannins characterized in Section 3.1 were distributed in independent bottles equipped with oxygen sensors and were oxidized for four consecutive air saturation cycles in duplicate.

As shown in Figure 1, four oxygen saturations were applied. The experiment lasted 52 days, and after each cycle the saturation times were prolonged: the first saturation lasted 9 days, the second 12 days, the third 14 days and the fourth 17 days. This behavior was observed for all the experimental samples, without significant differences regardless of the added tannins.

The total consumption of oxygen for all samples was around 23 mg/L of oxygen (Figure 2A). The addition of different commercial tannins in wines influenced the oxygen consumption rate on the 1st day (OCR 1°days). TT determined a higher consumption of oxygen likely due to its higher content of low molecular weight condensed tannins (Table 1). The initial OCR at the beginning of the first saturation was the fastest compared to the other cycles (Figure 2B), as already reported by Gambuti, Picariello, Rinaldi and Moio [24]. The TT sample showed values higher than the other samples with an increase of around 112% compared to the control. However, CT and ET also showed higher values compared to the control with an increase respectively of about 43% and 35%; only the GT sample did not show any variation compared to the control during the first saturation cycle. 

However, the trend of the total oxygen consumption rate for all of the saturation cycles did not show great differences among the samples (Figure 2C), and only the TT sample showed a significantly higher value than the other experimental samples during the first saturation. As shown in Figure 2, the speed of oxygen consumption of wines slows down as the cycles of saturation increase, probably because the main regulator in the speed of consumption of oxygen is the amount of sulfur dioxide, which converts the quinone back into the *o*-diphenol. When the amount of SO_2_ starts to decrease, the consumption of oxygen is less favored [12,25,26]. As reported by Vignault et al. [27], different behaviors among commercial tannins were observed, but while the authors detected higher total oxygen consumption rates when gallotannins and ellagitannins were added, in our study differences were not so relevant, probably because we resorted to real wines and not to model solutions used in previous works.

As expected, the exposure to oxygen determined a contemporary consumption of sulfur dioxide (Figure 3A,B). All the experimental samples at the beginning of the experiment showed values of total sulfur dioxide of around 66.56 ± 0.20 mg/L and values of free sulfur dioxide of 32.92 ± 0.33 mg/L. As shown in Figure 3, at the end of the third saturation, the values of both free and total sulfur dioxide were close to zero and, thus, it was not possible to measure them. Therefore, the addition of approximately 16 mg/L of oxygen during the first three saturations resulted in the loss of approximately 66 mg/L of SO_2_.

This theoretical value of consumed SO_2_ was determined considering a simplified model of wine oxidation, in which for each mole of oxygen, in the presence of metals, one mole of phenolic compound is oxidized to the corresponding quinone while one mole of hydrogen peroxide is produced [13]. Therefore, a O_2_:SO_2_ molar reaction ratio of 1:2, corresponding to a concentration ratio of 1:4 mg/L is usually considered for the estimation of SO_2_ consumption during wine oxidation [25]. However, a deviation from this theoretical value has already been detected during bottle aging [24] because it is likely that the effectiveness of this consumption depends on the flow of oxygen uptake and wine composition. 

During the three saturations it was possible to determine the consumption ratio between total sulfur dioxide and oxygen (Figure 3C). The sulfur dioxide consumed ranged from a minimum of 3.32 ± 0.19 mg / L for TT sample during the first oxygen saturations and a maximum of 4.62 ± 0.11 mg/L of the TT sample during the second oxygen saturation. However, it is not possible to exactly define the oxygen value that caused the loss of sulfur dioxide, since at the end of the third saturation, the sulfur dioxide values were not detectable.

### 2.3. Effects of Wine Oxygen Saturations on Acetaldehyde

A key parameter for monitoring wine aging is acetaldehyde, because it is expected to increase as a result of continuous oxygen exposure due to non-enzymatic oxidation of ethanol [16]. At the end of the experiment, the acetaldehyde amount had slightly increased (Figure 4), implying two important conclusions: 1) in agreement with Sheridan and Elias [28], when sulfur dioxide levels decrease, the bound form is released, and 2) when SO_2_ is consumed, acetaldehyde (Figure 4) starts to be produced by the Fenton reaction [29]. At the end of the experiment, the increase in acetaldehyde in the samples was variable: control samples and ET showed an increase of about 22% and 27%, respectively, while GT and CT showed the smallest increase of 18% and 12%, respectively. 

Although TT consumed a higher amount of oxygen, no significant production of acetaldehyde was detected. However, the values reported may not correspond to the actual acetaldehyde produced during the experiment. Indeed, according to Atanasova, Fulcrand, Cheynier and Moutounet [30] acetaldehyde may have been involved in the formation of ethyl-bridged anthocyanins and flavanols, as initially proposed by Timberlake and Bridle [31]. The reaction of acetaldehyde with malvidin and flavanols could be favored when condensed flavanols with high molecular weight (Table 1) are present, as for CT.

### 2.4. Effects of Different Enological Tannins on Wine Phenolics and Color Parameters during Oxygen Saturations

The possible involvement of acetaldehyde in the formation of new polymerized pigments is also confirmed by the variation of monomeric anthocyanins displayed in Table 2, which was already observed when the wine was exposed to oxygen supply [24]. The monomeric native anthocyanins drastically decreased during oxidative stress and the decrease was higher in wines added with tannins. 

Simultaneously, an increase in short polymeric pigments determined by the Harbertson method [16] was observed in all samples, with ET and GT being the ones showing the highest increase in these important stable compounds. The involvement of native anthocyanins in reactions yielding new polymeric pigments is consistent with the decrease in native anthocyanins shown in Table 2, and with similar effects observed in red wines during micro oxygenation [29,32].

Our results confirm that, through moderate oxygen exposure, these polymeric pigments become of crucial importance for wine color [33] and that small amounts of acetaldehyde can react with anthocyanins to produce new stable red pigments, as reported by different authors [31,34].

In agreement with recent studies, differences in terms of color intensity were detected (Table 2) with positive effects determined by tannin addition probably due to the copigmentation phenomenon [27] and the contemporary formation of new stable pigments [17]. Indeed, the color intensity increased for all the samples during the oxygen treatment, consistently with the findings of Atanasova, Fulcrand, Cheynier and Moutounet [30] showing that micro-oxygenated wines have a higher C.I. than untreated ones. Nevertheless, the control sample showed a lower increase than the samples spiked with tannins likely due to the great influence of new pigments involving enological tannins and/or acetaldehyde. Treated samples also showed a slight increase in hue (yellow tint) due to an increase in the abs at 420 nm owing to the effect of oxidative stress in wines. As reported by Somers [35], an increase in abs at 420 nm due to the formation of yellow pigments during wine maturation occurred.

The amount of total phenols determined by the Adam assay and the flavans reactive to vanillin did not show a statistical difference after the additions of the different classes of tannins (Table 3). Furthermore, a comparison between the wine before the oxygen supply and that at the end of the experiment showed that the amount of total phenols was higher in the treated wines, probably because of a higher reactivity towards the reactants used for the analysis of the new compounds formed in wines after the tannin addition and the oxygen supply. A great loss of vanillin reactive flavans with respect to the control was instead detected for CT, showing that polymerization increased, as already observed during micro oxygenation [29].

### 2.5. Effects of Different Enological Tannins on SPI and Astringency Subqualities at the End of the Experiment

As wine tannins are heterogeneous mixtures containing a range of polymers with different sizes, subunit compositions, and subunit linkages, and as a complete analysis of all phenolic composition is difficult to realize, the analysis of protein precipitation after the reaction with wine is an indirect simple measure of potential astringency. The most accredited mechanism for astringency involves the interaction between wine tannins and saliva proteins [36]. An in vitro measurement of the astringency by the Saliva Precipitation Index (SPI) was performed on wines before (zero time) and at the end of the experiment, as shown in Figure 5A. An increase in the precipitation of salivary proteins was observed in TT wine with respect to the control wine. The addition of the TT highly influenced the formation of interactions and the binding with saliva. The commercial TT contained mainly epicatechin, epigallocatechin and epigallocatechin gallate, as well as A-type and B-type proanthocyanidins. The degree of polymerization of proanthocyanidins reached pentamers and presented a high degree of galloylation, which may be responsible for the high binding with salivary proteins. EGCG, which is a major component of TT, can bind in multi-dentate fashion to multiple sites on the protein surface [37], enhancing the saliva precipitation. For this reason, TT does not have a positive effect in reducing the potential astringency, not even when slight oxidation occurred. In fact, after saturation, the SPIs of TT and control wines were higher than the other wines. Surprisingly, before saturation, there was no difference in reactivity of ET, CT and GT towards salivary proteins even if the composition of these commercial tannins was different. The ET was characterized mainly by ellagitannins, CT contained up to hexameric polymers with a very low level of galloylation, while GT was composed of quinic acid with different gallate subunits (Table 1). The presence of carbohydrates including glucose and fructose in these tannins may have interfered with the binding of salivary proteins to the tannin molecules [38]. However, the addition of CT and ET tannins, after exposure to oxygen, resulted in wines with lower potential astringency, of about 4 g/L of GAE with respect to 6 g/L of control wines. The oxygen saturation cycles had a significant effect (*p* < 0.05) in reducing the precipitation of salivary proteins and then the astringency of wine. In a previous work, we observed that oxygen can enhance the formation of polymeric pigment between ellagitannins (ET) and wine polyphenols, thus increasing the color stability and reducing the precipitation of salivary proteins [17]. A similar behavior was also observed in different Italian wines after one year of aging with ET and CT tannins [19].

The addition of enological tannins may also influence the sensory properties of red wines, and in particular the effect of oxygen exposure on wine astringency subqualities has not been studied yet. A trained jury evaluated wines for the qualitative astringency using the CATA questionnaire with a list of subqualities (silk, velvet, dry, corduroy, adhesive, aggressive, hard, soft, mouthcoat, rich, full-body, green, grainy, satin, pucker, persistent) from which the panelist has to select the sensation they consider appropriate for the tasted wine. The subqualities characterizing the wines after saturation were expressed as frequency citations (Cf%). Figure 5B showed the contribution of each wine on the total Cf% within each attribute. 

The control wine was characterized by the terms green and silk, indicating that after saturation, a high acidity associated with astringency (green) was perceived. The ET wine was characterized as grainy and velvet, and the latter sensation seemed to be due to the formation of pigmented polymers, which have been previously correlated with the velvety subquality [19]. The TT wine was characterized by the negative attributes of astringency, such as dry and adhesive, highly correlated with a high tannin content [14], also in accordance with higher values of SPI. However, the benefit of low oxygen exposure in TT wine was in the corduroy term that can be associated to the sensation of a light wrinkling of the soft palate that can be felt by tongue movements, and to the full-body sensation. While TT should have some positive effects on wine after saturation, the negative ones prevailed. The GT wine was highly defined as dry, pucker and hard, so that the addition of gallotannins increased the bitter sensation associated with a strong astringency. CT addition conferred to wine adhesive, grainy, velvety, soft, and satin sensations. In particular, CT wine was highly characterized by positive astringency subqualities (velvet, soft, satin), probably due to the formation of new compounds between polyphenols and oxidation products.

## 3. Materials and Methods

### 3.1. Experimental Wines

The red wine was produced during the 2018 vintage with 100% Aglianico at Tenute Casoli winery (Candida, Avellino, Italy). After the end of the malolactic fermentation, the wine was microfiltered (0.45 micron) to remove residual yeasts to avoid any microbial involvement in the process. The base parameters were the following: alcohol 13.70 ± 0.12% *v*/*v*, malic acid 0.1 ± 0.05 g/L, tartaric acid 5.75 ± 0.1, pH 3.38 ± 0.05 volatile acidity 0.47 ± 0.08 g/L

A control wine and four additional wines were prepared by separately adding each of them with a different type of tannins (30 g/hL). The treated wines were contained in a 5 L flask previously filled with the Aglianico wine and saturated with nitrogen. The four types of tannins were: gallotannins (GT) (Galalcool, Laffort Oenologie, Bordeaux, France), condensed tannins (CT) (Tannin VR Grape, Laffort Oenologie, Bordeaux, France), ellagitannins (ET) (Quertanin, Laffort Oenologie, Bordeaux, France) and tea tannins (TT) (Silviachimica, Cuneo, Italy).

Samples were oxidized by air saturation. For each wine, two separate saturation replicates (750 mL each) were carried out until the oxygen level of the wine reached 6.0 mg/L. This was done by gently shaking 4.5 L of wine in a 5 L closed flask, then the cap was opened to allow fresh air to enter, and the shaking operation was repeated 2 more times [39]. Four bottles in duplicate were prepared and each of them was analyzed at the end of each saturation.

Wines were stored in an incubator in the dark at 20 °C and dissolved oxygen level was monitored once a day with a Nomasense oxygen analyzer from Nomacorc S.A. (Thimister-Clermont, Belgium). The legend of the different acronyms was reported in Table 4

### 3.2. Aglianico Wine

The Aglianico cultivar includes three biotypes, Taurasi, Taburno and Vulture, all grown in southern Italy, mainly in the regions of Campania and Basilicata.

Several studies report that at technological maturity, Aglianico grapes were characterized by sugar levels generally ranging from 220 to 240 g/L and pH values between 3.10 and 3.40 [40,41]. Aglianico grape shows a good phenolic maturity [42] and related wines are rich in phenolic compounds, mainly anthocyanins and tannins [43]. Regarding the tannins, the ones extracted from grape seeds of Aglianico, compared to those of the skins, were more reactive towards salivary proteins [42,44]. A past study of Rinaldi, Gambuti, and Moio, [45] reported that the tannins of the grape seeds compared to those of the skins are characterized by a high percentage of galloylation, a medium-low polymerization degree and the absence of epigallo-catechin in proanthocyanidins. Concerning aroma, despite the absence of specific molecular markers in Aglianico wine, it was possible to easily differentiate it from Merlot and Cabernet Sauvignon wines [46]. The molecules ethyl butanoate, ethyl-2-methylbutanoate, ethyl-3-methylbutanoate and ethyl-2-methylpropanoate appeared to be involved in its characteristic odor of red berry fruits.

### 3.3. Determination of Sulfur Dioxide

Free and total sulfur dioxide were both determined by the official method of analysis (Compendium of International Methods of Wine and Must Analysis 2019).

### 3.4. High-Performance Liquid Chromatography Analysis of Acetaldehyde

Acetaldehyde was determined by HPLC after the derivatization reaction with 2,4-dinitrophenylhydrazine reagent (Aldrich chemistry, Saint Louis, MO, USA)) as reported by Han et al. [47]. Briefly, wine sample aliquots (100 μL) were dispensed into a vial, followed by the addition of 20 μL of freshly prepared 1.120 mg/L SO_2_ solution, 20 μL of 25% sulfuric acid (Carlo Erba reagent 96%), and 140 μL of 2 g/L 2.4-dinitrophenylhydrazine reagent. After mixing, the solution was allowed to react for 15 min at 65 °C and then promptly cooled to room temperature. Carbonyl hydrazones were analyzed by HPLC using a HPLC Shimadzu LC10 ADVP apparatus (Shimadzu, Milan, Italy,), consisting of a SCL-10AVP system controller, two LC-10ADVP pumps, an SPD-M 10 AVP detector (Shimadzu, Kyoto, Japan), and an injection system full Rheodyne model 7725 (Rheodyne, Cotati, CA, USA) equipped with a 50 μL loop. The separation was carried out in a Waters Spherisorb column (250 × 4.6 mm, 4 μm particles diameter) equipped with a guard column. Optimum efficiency of separation was obtained by using a flow rate of 0.75 mL/min, column temperature of 35 °C; mobile phase solvents were: (A) 0.5% formic acid (Sigma Aldrich ≥ 95%, Saint Louis, MO, USA) in water milli-Q (Sigma Aldrich) and (B) acetonitrile (Sigma Aldrich ≥ 99.9%); gradient elution protocol was: 35% B to 60% B (t = 8 min), 60% B to 90% B (t = 13 min), 90% B to 95% B (t = 15 min, 2-min hold), 95% B to 35% B (t = 17 min, 4-min hold), total run time, 21 min. Eluted peaks were compared with derivatized acetaldehyde standard.

### 3.5. High-Performance Liquid Chromatography Analyses of Anthocyanins

Separation of monomeric anthocyanins was carried out as reported in the Compendium of International Methods of Wine and Must Analysis by using a HPLC system previously described equipped with a column heating device set at 40 °C, with a C18 column, Waters Spherisorb column (250 × 4.6 mm, 4 μm particles diameter) with pre-column. All the samples were filtered through 0.45mm Durapore membrane filters (Millipore, Cork, Ireland) into glass vials and immediately injected into the HPLC system. A 50 μL loop was used. Briefly, elution was carried out by using a flow rate of 0.80 mL/min. Eluents were: solvent A consisting of water milli-Q (Sigma Aldrich)/formic acid (Sigma Aldrich ≥ 95%)/acetonitrile (Sigma Aldrich ≥ 99.9%) (87:10:3) *v*/*v* and, solvent B consisting of water/formic acid/acetonitrile (40:10:50) *v*/*v*. The following gradient was used: zero-time conditions 94% A and 6% B, after 15 min the pumps were adjusted to 70% A and 30% B, at 30 min to 50% A and 50% B, at 35 min to 40% A, and 60% B, at 41 min, end of analysis, to 94% A and 6% B. After a 10 min equilibrium period, the next sample was injected. For calibration, the external standard method was used: the calibration curve was plotted for the malvidin-3-*O*-glucoside (Extrasynthese, Lyon, France) on the basis of peak area. The concentration of the following monomeric anthocyanins was determined: delphinidin 3-*O*-glucoside, cyanidin 3-*O*-glucoside, peonidin 3-*O*-glucoside, malvidin 3-*O*-glucoside, and malvidin 3-(6′′-acetyl)-*O*-glucoside. The concentration was expressed as mg/L of malvidin-3-*O*-glucoside. The calibration curve was characterized by a correlation coefficient (r^2^) > 0.998. The LOD was determined as a signal-to-noise ratio (S/N) of 3:1 (0.0262 mg L^−1^), and the LOQ was determined as a S/N of 10:1 (0.0866 mg L^−1^).

### 3.6. Analysis of the Chromatic Characteristics and Phenolic Compounds of the Wine

The color and the phenolic compound content were determined as they are responsible for the fundamental organoleptic properties of wines such as visual, astringency and bitterness. The wine color analysis was performed by spectrophotometric techniques (color intensity and hue). The short polymeric pigments (SPP), and total phenols were determined by the Harbertson–Adams assay [48]. Vanillin reactive flavans (VRF) were determined as previously reported [17]. All analyses were conducted by two experimental replicas and two analytical replicas.

### 3.7. The Saliva Precipitation Index

The SPI, defined as the reactivity of wine tannins towards salivary proteins, is used to measure astringency as it is well correlated to the perceived intensity of the overall sensation [45]. Salivary proteins were obtained by mixing resting saliva samples from six non-smoking volunteers (three males and three females). Saliva collection was performed between 10 and 11 a.m. The resulting mix was centrifuged at 10,000 *g* for 10 min to remove any insoluble material, and the supernatant was used for the analyses. The binding reaction between wine and saliva was performed at 37 °C, in-mouth temperature, for 5 min. After this period, wine tannins bound to proteins were separated from supernatant by centrifugation (10 min 10,000 *g* at 4 °C), which was analyzed by chip electrophoresis. The commercial Experion Pro260 analysis kit (Bio-Rad, Milano, Italy) and Experion system were used for the SPI determination, as described by Rinaldi et al., 2014 [49]. The SPI was calculated by the percentage reduction in the fluorescence signal of salivary proteins with respect to control saliva (before reaction). The reduction in salivary bands after the binding reaction with standard solutions at different tannin concentrations was plotted against the concentration of the same solutions measured by Folin–Ciocalteu method and expressed as GAE, and the generated best-fit polynomial was used to calculate the SPI. SPI was performed twice for each treatment, and results represent the mean ± standard deviation of four replicates, expressed as mg/L of gallic acid equivalent (GAE).

### 3.8. Wine Evaluation

Thirteen judges from the Division of Sciences of Vine and Wine, Department of Agriculture, University of Naples Federico II, in Avellino (Italy), trained for astringency and mouthfeel sensations [19], participated in wine evaluation sessions. In each session were performed two tasting evaluations of four unknown samples. Each session comprised two tasting evaluations of four unknown samples. These were presented in balanced random order at room temperature (18 ± 2 °C) in black tulip-shaped glasses coded with three-digit random numbers. The in-mouthfeel sensations were evaluated by using the method described by Rinaldi and Moio [19]. After rating the overall astringency (maximum of the perceived intensity), the judges used the CATA question with the describing astringency subqualities (silk, velvet, dry, corduroy, adhesive, aggressive, hard, soft, mouthcoat, rich, full-body, green, grainy, satin, pucker, persistent), checked the subqualities (if any), and then rated the intensity of the sensation (rate-all-that-apply, RATA).

### 3.9. NMR Experiments

^1^H-NMR experiments were run on Varian Unity Inova 700 spectrometer (Varian, Palo Alto, CA, USA) equipped with a 13C Enhanced HCN Cold Probe and by using a Shigemi 5 mm NMR tubes. Standard Varian pulse sequences were employed. Two milligrams of each sample were solubilized in 600 μL of CD_3_OD as a deuterated solvent.

### 3.10. MS Experiments

Mass Spectrometric studies were performed by using the linear ion trap LTQ Orbitrap XL hybrid Fourier Transform MS (FTMS) instrument equipped with an ESI ION MAX source (Thermo-Fisher, Waltham, MA, USA) and coupled to an Agilent 1100 LC binary system including a solvent reservoir, online degasser, binary pump, and thermostated autosampler. All the solvents and reagents used were of laboratory grade. The following source settings were used for HR-MS: spray voltage 4.5 kV, capillary temperature 350 °C, capillary voltage 30 V, sheath gas 20 and auxiliary gas 21 (arbitrary units), and tube lens voltage 60 V. The LOD was determined as a signal-to-noise ratio (S/N) of 3:1 (30 ng mL^−1^ for catechin, and 40 ng mL^−1^ for epicatechin).

### 3.11. Data Analysis

Sensory attributes were evaluated by Duncan test over two replicates. Differences of *p* < 0.05 were considered significant. Elaborations were carried out by means of XLSTAT software (Addinsoft, XLSTAT 2017).

## 4. Conclusions

This study provides clear evidence that the addition of different classes of tannins and the final quality of wines after oxygen supply are entwined. The results show that in a complex matrix, such as young red wines, the addition (30 g/hL) of different types of commercial tannins differently influenced the oxygen consumption on the 1st day, the amount of acetaldehyde, the native anthocyanins, the potential astringency (SPI) and wine astringency subqualities.

The results suggest that, as different enological tannins have different impacts on wine evolution under oxygen exposure, they need to be selected on the basis of the winemaker’s aim. TT are more useful to favor the fast consumption of oxygen, e.g., during the first phases of winemaking of red wines less rich in polyphenols, while ET and CT better serve the purpose of improving astringency during the aging of wines rich in polyphenols, such as Aglianico wines. Further studies on red wines with different initial phenolic composition could help to better finalize the use of different enological tannins.

## Figures and Tables

**Figure 1 molecules-25-04607-f001:**
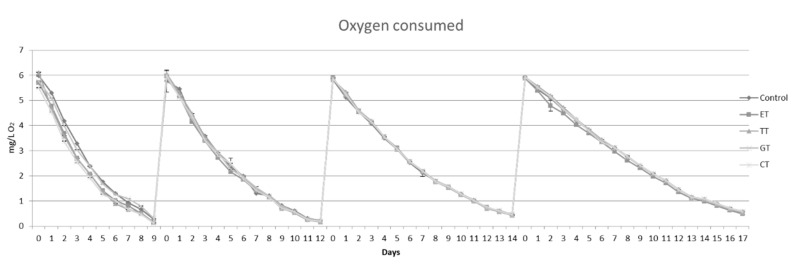
Average oxygen concentrations (mg/L) measured in each wine. All the data are expressed as means ± standard deviation.

**Figure 2 molecules-25-04607-f002:**
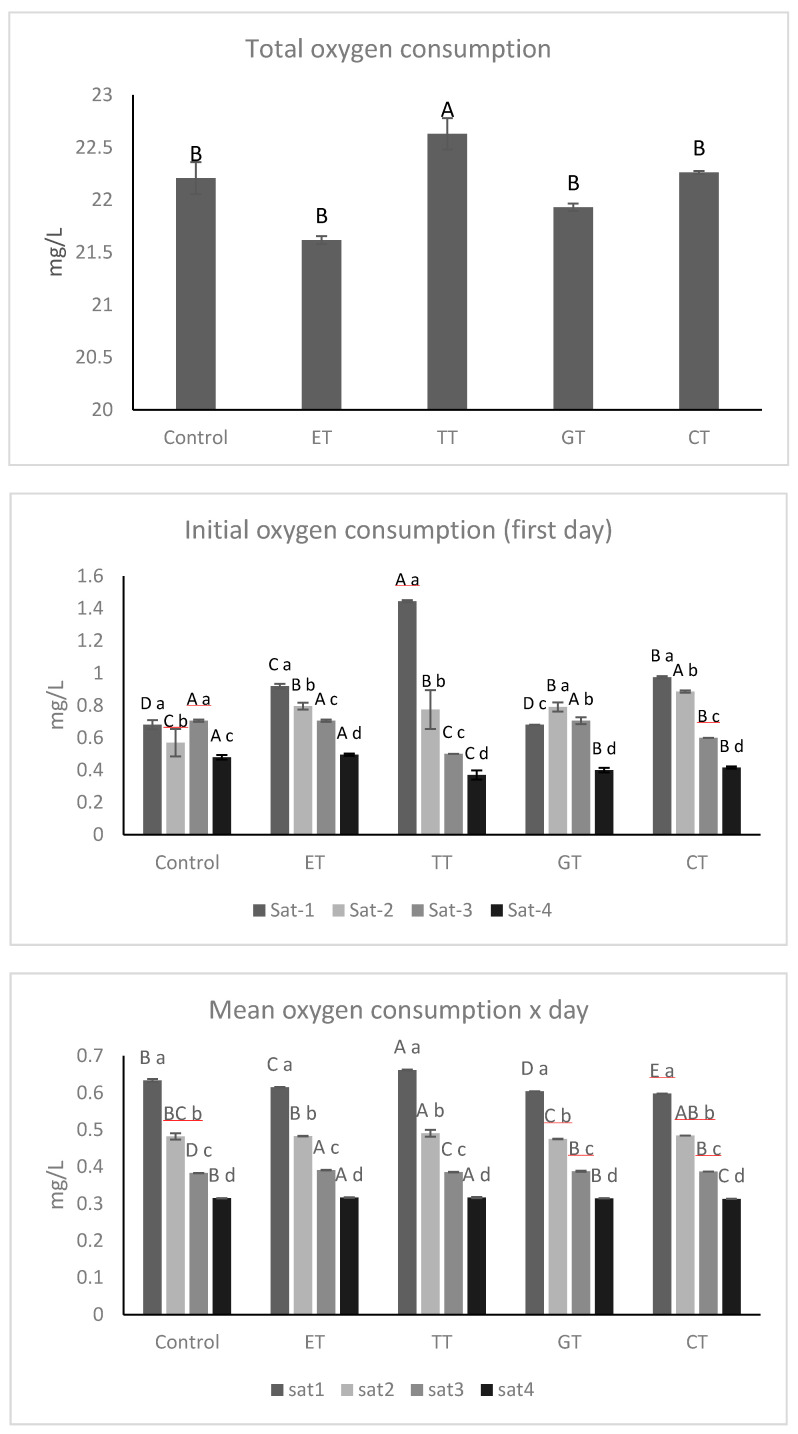
(**A**) Total oxygen consumption; (**B**) initial oxygen consumption rate on the first day; (**C**) mean oxygen consumption rate per day for each saturation cycle. All the data are expressed as means ± standard deviation. Different letters indicate a statistically significant difference among treated wines during each saturation time (A, B, C, D, E) and among treated wines for the same saturation time (a, b, c, d). All the data are expressed as means ± standard deviation, (*p* < 0.05).

**Figure 3 molecules-25-04607-f003:**
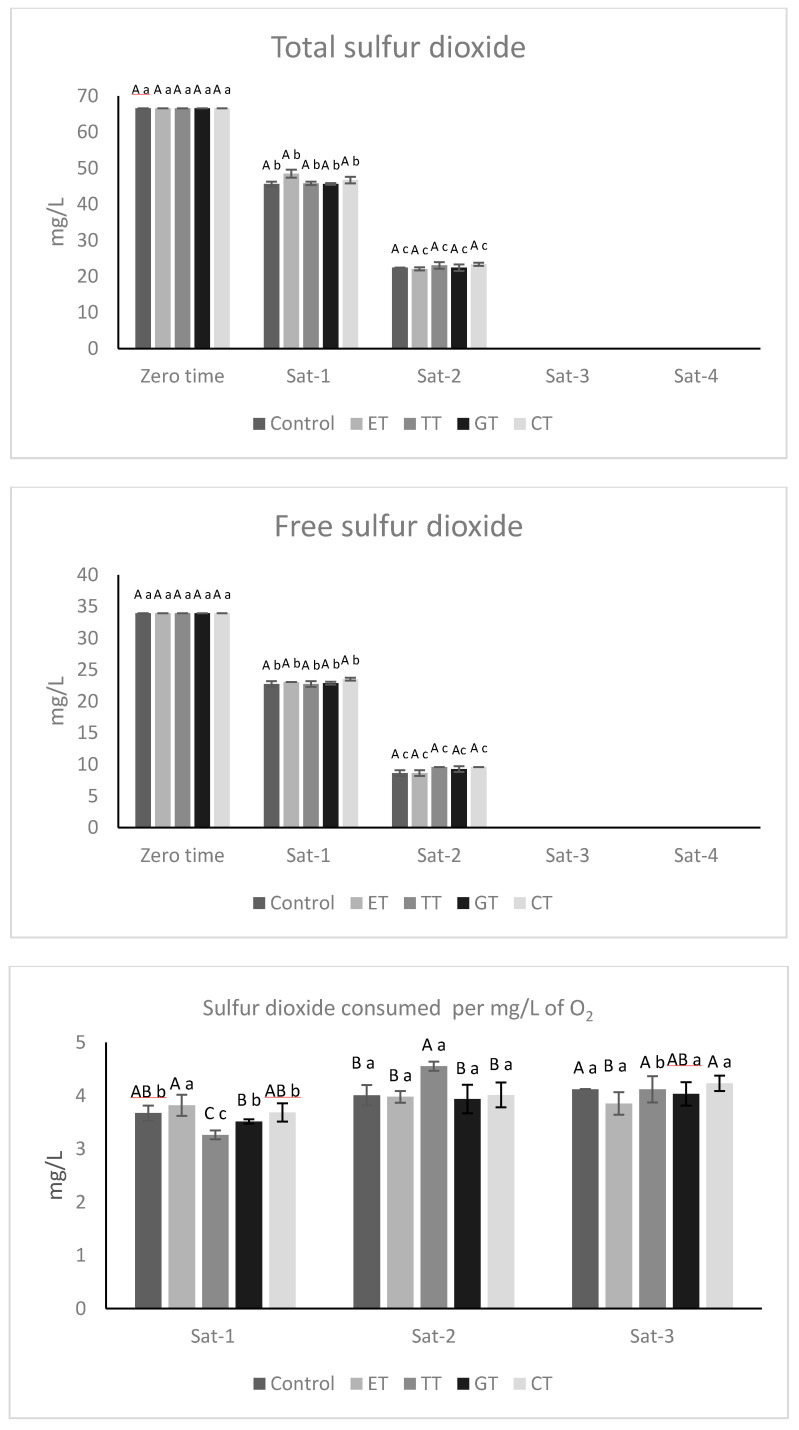
(**A**) Evolution of free sulfur dioxide and (**B**) evolution of total sulfur dioxide. (**C**) Sulfur dioxide consumed per mg/L of oxygen. All the data are expressed as means ± standard deviation. Different letters indicate a statistically significant difference among treated wines during each saturation time (A, B, C, D) and among treated wines for the same saturation time (a, b, c, d). All the data are expressed as means ± standard deviation, (*p* < 0.05).

**Figure 4 molecules-25-04607-f004:**
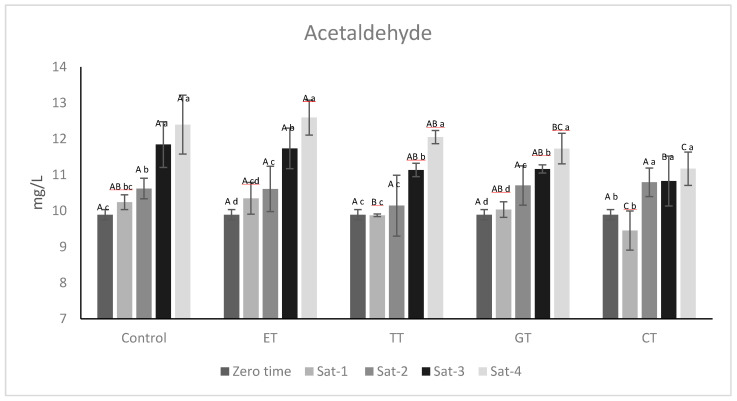
Evolution of acetaldehyde. All the data are expressed as means ± standard deviation. Different letters indicate a statistically significant difference among treated wines during each saturation time (A, B, C, D) and among treated wines for the same saturation time (a, b, c, d). All the data are expressed as means ± standard deviation, (*p* < 0.05).

**Figure 5 molecules-25-04607-f005:**
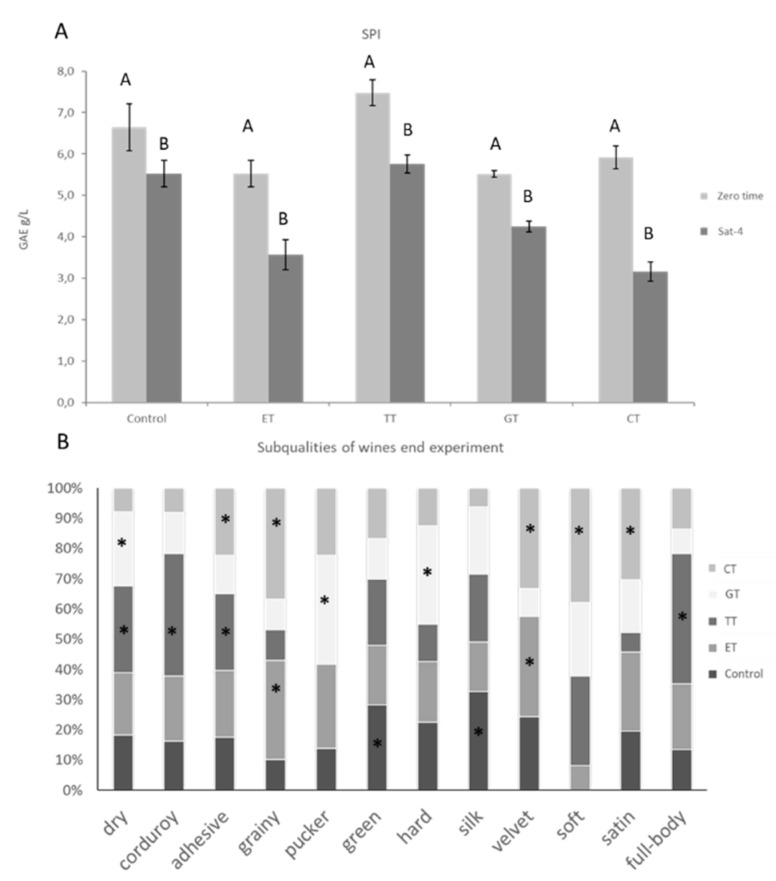
(**A**) Saliva Precipitation Index (SPI, g/L of gallic acid equivalent, GAE), (**B**) subqualities of wines at the end of the experiment. Different letters indicate a statistically significant difference in SPI before and after the oxidation oxygen treatment (A, B), while for a different stage of oxygen saturations (a, b, c) differences among subqualities of wines at the end experiment are denoted with an asterisk (*p* < 0.05).

**Table 1 molecules-25-04607-t001:** Monomers and oligomers identified by HR ESI-MS (negative ion mode) in the ellagitannin (ET), tea tannin (TT), gallotannin (GT), and condensed tannin (CT) mixtures. Identification of compounds was proposed on the basis of the molecular formula implied by the ion peak values with a mass tolerance of 10 ppm; RDB is the degree of unsaturation represented as rings and/or double bonds.

ET-Mixture		TT-Mixture		GT-Mixture		CT-Mixture					
*m/z* Formula RDB	Compd	*m/z* Formula RDB	Compd	*m/z* Formula RDB	Compd	*m/z* Formula RDB	Compd	*m/z* Formula RDB	Compd	*m/z* Formula RDB^a^	Compd
169.0132 C_7_H_5_O_5_ RDB = 5	Gallic acid	289.0697 C_15_H_13_O_6_ RDB=9	EC monomer	169.0133 C_7_H_5_O_5_ RDB=5	Gallic acid (G)	289.0697 C_15_H_13_O_6_ RDB=9	(E)C monomer	865.1932 C_45_H_37_O_18_ RDB=27	3 (E)C trimer	1451.2636 C_75_H_55_O_31_ RDB=48	4 (E)C + (E)GC pentamer (3 A-type bonds)
300.9968 C_14_H_5_O_8_ RDB = 12	Ellagic acid	305.0645 C_15_H_13_O_7_ RDB=9	EGC monomer	191.0549 C_7_H_11_O_6_ RDB=2	Quinic acid (Q)	305.0644 C_15_H_13_O_7_ RDB=9	(E)GC monomer	877.1555 C_45_H_33_O_19_ RDB=29	2 (E)C + (E)GC trimer (2 A-type bonds)	1453.2787 C_75_H_57_O_31_ RDB=47	4 (E)C + (E)GC pentamer (2 A-type bonds)
481.0586 C_20_H_17_O_14_ RDB = 12	Hexahydroxy diphenoyl-glucose	441.0794 C_22_H_17_O_10_ RDB=14	ECG monomer	343.0646 C_14_H_15_O_10_ RDB=7	1 Q + 1 G	441.0794 C_22_H_17_O_10_ RDB=14	(E)CG monomer	879.1717 C_45_H_35_O_19_ RDB=28	2 (E)C + (E)GC trimer (1 A-type bond)	1455.2912 C_75_H_59_O_31_ RDB=46	4 (E)C + (E)GC pentamer (1 A-type bond)
631.0528 C_27_H_19_O_18_ RDB = 18	Castalin	457.0742 C_22_H_17_O_11_ RDB=14	EGCG monomer	495.0742 C_21_H_19_O_14_ RDB=12	1 Q + 2 G	457.0741 C_22_H_17_O_11_ RDB=14	(E)GCG monomer	1147.2048 C_60_H_43_O_24_ RDB=39	4 (E)C tetramer (3 A-type bonds)	1723.3264 C_90_H_67_O_36_ RDB=57	6 (E)C hexamer (3 A-type bonds)
633.0658 C_27_H_21_O_18_ RDB = 17	Corilagin	591.0946 C_30_H_23_O_13_ RDB=19	EC + EGC dimer (1 A-type bond)	647.0840 C_28_H_23_O_18_ RDB=17	1 Q + 3 G	575.1150 C_30_H_23_O_12_ RDB=19	2 (E)C dimer (1 A-type bond)	1149.2216 C_60_H_45_O_24_ RDB=38	4 (E)C tetramer (2 A-type bonds)	1725.3431 C_90_H_69_O_36_ RDB=56	6 (E)C hexamer (2 A-type bonds)
635.0837 C_27_H_23_O_18_ RDB = 16	Tri-*O*-galloyl-glucose	607.0893 C_30_H_23_O_14_ RDB=19	2 EGC dimer (1 A-type bond)	799.0937 C_35_H_27_O_22_ RDB=22	1 Q + 4 G	577.1306 C_30_H_25_O_12_ RDB=19	2 (E)C dimer	1151.2376 C_60_H_47_O_24_ RDB=37	4 (E)C tetramer (1 A-type bond)	1727.3500 C_90_H_71_O_36_ RDB=55	6 (E)C hexamer (1 A-type bond)
783.0625 C_34_H_23_O_22_ RDB = 23	Pedunculagin	647.0839 C_30_H_24_KO_14_ RDB=18	2 EGC dimer	951.1036 C_42_H_31_O_26_ RDB=27	1 Q + 5 G	591.0946 C_30_H_23_O_13_ RDB=19	(E)C + (E)GC dimer (1 A-type bond)	1153.2532 C_60_H_47_O_24_ RDB=36	4 (E)C tetramer	1729.3663 C_90_H_73_O_36_ RDB=54	6 (E)C hexamer
785.0763 C_34_H_25_O_22_ RDB = 22	Tellimagrandin I	799.0936 C_37_H_28_KO_18_ RDB=23	EGC + EGCG dimer	1103.1134 C_49_H_35_O_30_ RDB=32	1 Q + 6 G	607.0893 C_30_H_23_O_14_ RDB=19	2 (E)GC dimer (1 A-type bond)	1165.2165 C_60_H_45_O_25_ RDB=38	3 (E)C + (E)GC tetramer (2 A-type bonds)	1739.3183 C_90_H_67_O_37_ RDB=57	5 (E)C + (E)GC hexamer (3 A-type bonds)
933.0565 C_41_H_25_O_26_ RDB = 29	Castalagin and/or Vescalagin	951.1034 C_45_H_36_KO_21_ RDB=27	3 EGC trimer	1255.1230 C_56_H_39_O_34_ RDB=37	1 Q + 7 G	647.0839 C_30_H_24_KO_14_ RDB=18	2 (E)GC dimer	1435.2684 C_75_H_55_O_30_ RDB=48	5 (E)C pentamer (3 A-type bonds)	1741.3300 C_90_H_69_O_37_ RDB=56	5 (E)C + (E)GC hexamer (2 A-type bonds)
1065.0978 C_46_H_33_O_30_ RDB = 30	Grandinin	1103.1133 C_52_H_40_KO_25_ RDB=32	2 EGC + EGCG trimer			861.1605 C_45_H_33_O_18_ RDB=29	3 (E)C trimer (2 A-type bonds)	1437.2822 C_75_H_57_O_30_ RDB=47	5 (E)C pentamer (2 A-type bonds)	1743.3071 C_90_H_71_O_37_ RDB=55	5 (E)C + (E)GC hexamer (1 A-type bond)
						863.1763 C_45_H_35_O_18_ RDB=28	3 (E)C trimer (1 A-type bond)	1439.2955 C_75_H_59_O_30_ RDB=46	5 (E)C pentamer (1 A-type bond)		

Epigallocatechin-3-*O*-gallate (EGCG), epigallocatechin (EGC), gallocatechin-3-*O*-gallate (GCG), gallocatechin (GC), catechin (C), epicatechin (EC), (epi) gallocatechin (E)GC and (epi)catechin-3-*O*-gallate ((E)CG).

**Table 2 molecules-25-04607-t002:** Native anthocyanins, polymeric pigments and chromatic characteristics.

	Samples	Total Anthocyanins mg/L	Short Polymeric PigmentsAbs	Color IntensityAbs	Tonality
Zero Time	Control	1795.13 ± 17.37 A a	0.28 ± 0.01 A b	6.84 ± 0.05 B e	0.51 ± 0.00 ABC c
	ET	1795.13 ± 17.37 A a	0.28 ± 0.00 A c	6.82 ± 0.00 BC e	0.51 ± 0.00 C d
	TT	1795.13 ± 17.37 A a	0.28 ± 0.00 A d	6.77 ± 0.01 C e	0.51 ± 0.00 BC d
	GT	1795.13 ± 17.37 A a	0.28 ± 0.02 A d	6.82 ± 0.04 BC e	0.52 ± 0.01 AB c
	CT	1795.13 ± 17.37 A a	0.28 ± 0.00 A c	6.94 ± 0.01 A e	0.52 ± 0.00 A d
Sat.-1 (9d)	Control	1715.06 ± 74.88 BC b	0.26 ± 0.00 B c	7.32 ± 0.09 A d	0.51 ± 0.00 B c
	ET	1681.19 ± 29.41 C b	0.27 ± 0.00 A d	7.39 ± 0.06 A d	0.52 ± 0.00 A c
	TT	1758.16 ± 33.91 AB a	0.26 ± 0.00 B e	7.35 ± 0.11 A d	0.51 ± 0.00 B d
	GT	1813.51 ± 44.49 A a	0.27 ± 0.00 A e	7.35 ± 0.03 A d	0.51 ± 0.00 B c
	CT	1741.04 ± 35.88 BC a	0.27 ± 0.00 A d	7.37 ± 0.03 A d	0.51 ± 0.00 B e
Sat.-2 (12d)	Control	1679.00 ± 4.31 A b	0.28 ± 0.01 C b	7.79 ± 0.05 B c	0.51 ± 0.00 B c
	ET	1621.58 ± 45.35 A b	0.28 ± 0.00 C c	7.92 ± 0.00 A c	0.52 ± 0.00 A c
	TT	1672.76 ± 23.72 A b	0.30 ± 0.00 A c	7.96 ± 0.10 A c	0.52 ± 0.01 A c
	GT	1681.54 ± 3.96 A b	0.29 ± 0.00 B c	8.02 ± 0.07 A c	0.53 ± 0.00 A c
	CT	1532.24 ± 107.80 B b	0.29 ± 0.00 B b	8.00 ± 0.09 A c	0.52 ± 0.00 A c
Sat.-3 (14d)	Control	1517.78 ± 55.37 A c	0.31 ± 0.00 B a	8.70 ± 0.33 A a	0.56 ± 0.00 A b
	ET	1420.47 ± 64.09 B c	0.30 ± 0.01 B b	8.76 ± 0.09 A a	0.55 ± 0.00 A b
	TT	1428.70 ± 40.49 B c	0.31 ± 0.00 A b	8.88 ± 0.04 A a	0.55 ± 0.00 A b
	GT	1446.34 ± 0.75 B c	0.30 ± 0.00 C b	9.00 ± 0.46 A a	0.54 ± 0.02 A b
	CT	1467.64 ± 21.04 AB b	0.32 ± 0.00 A a	8.75 ± 0.06 A a	0.55 ± 0.00 A b
Sat.-4(17d)	Control	1168.75 ± 44.13 A d	0.32 ± 0.01 B a	8.14 ± 0.21 B b	0.60 ± 0.01 B a
	ET	1121.57 ± 29.36 BC d	0.33 ± 0.01 A a	8.58 ± 0.07 A b	0.60 ± 0.01 AB a
	TT	1133.60 ± 3.46 ABC d	0.34 ± 0.01 A a	8.53 ± 0.07 A b	0.61 ± 0.00 A a
	GT	1099.69 ± 1.86 C d	0.34 ± 0.01 A a	8.50 ± 0.01 A b	0.61 ± 0.00 A a
	CT	1160.10 ± 36.23 AB c	0.32 ± 0.00 B a	8.56 ± 0.11 A b	0.61 ± 0.00 A a

All the data are expressed as means ± standard deviation of four replicates (two experimental replicates per two analytical replicates). Different letters indicate a statistically significant difference between enological tannins additions during oxygen treatment (A, B, C), while for a different stage of oxygen saturations (a, b, c) (*p* < 0.05).

**Table 3 molecules-25-04607-t003:** Total phenols, flavans reactive to vanillin and tannins reactive to BSA.

	Samples	Total Phenols mg/L	Flavans Reactive to Vanillin mg/L
Zero Time	Control	2039.05 ± 33.41 A a	1090.89 ± 47.54 A a
	ET	2058.54 ± 88.33 A a	1101.11 ± 51.92 A a
	TT	2034.87 ± 18.40 A b	1112.89 ± 59.53 A a
	GT	2020.95 ± 58.11 A b	1099.53 ± 42.86 A a
	CT	2069.68 ± 43.25 A a	1094.04 ± 43.49 A a
Sat.-4	Control	2033.48 ± 89.21 B b	1062.61 ± 89.04 AB a
	ET	2121.18 ± 53.29 AB a	1049.26 ± 38.14 BC b
	TT	2151.81 ± 27.42 A a	1136.46 ± 18.39 A a
	GT	2126.75 ± 39.73 A a	1024.12 ± 48.34 BC b
	CT	2126.75 ± 67.03 A a	984.84 ± 18.59 C b

Different letters indicate a statistically significant difference among enological tannins additions during oxygen treatment (A, B, C), while for a different stage of oxygen saturations (a, b, c), if the same letter is present, no significant difference was detected (*p* < 0.05).

**Table 4 molecules-25-04607-t004:** Codes and description of experimental wines.

Code	Description
	Experimental wines
Control	Wine control
ET	Wine with addition of ellagitannin (30 g hL^−1^)
TT	Wine with addition of tea tannins (30 g hL^−1^)
GT	Wine with addition of gallotannins (30 g hL^−1^)
CT	Wine with addition of condensed (tannin 30 g hL^−1^)
	Saturations time
Zero Time	Zero analysis time
Sat.-1	First oxygen saturation
Sat.-2	Second oxygen saturation
Sat.-3	Third oxygen saturation
Sat.-4	Fourth oxygen saturation

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
