# Peer review of "Effect of Different Enological Tannins on Oxygen Consumption, Phenolic Compounds, Color and Astringency Evolution of Aglianico Wine"

_molecules, 2020, doi:10.3390/molecules25204607_

Round 1

Reviewer 1 Report

The paper entitled “Effect of different enological tannins on oxygen consumption, phenolic compounds, color and astringency evolution on Aglicano wine” reports the use of four different tannins in wine composition and sensory characteristics. It is an interesting paper and I think it is worth its publication although some improvements should be done before publication.

Abstract: The sentence “the astringency, determined before and at the end of the experiment, decreased in all the samples and mostly in the ET and GT samples” should be changed by “the astringency, determined before and at the end of the experiment, decreased in all the samples, included the control wine, and mostly in the ET and GT samples”

Introduction

Line 39: some mention to Tea tannins should be done

Line 57-62: A comment regarding the practical aspect of this study could be really appreciated.

Material and Methods

Line 68: malic acid

Line 120: Why Vitisin A, usually present at higher concentration than Vitisin B, was not quantified?

Results and discussion

Line 330-336 The formation of compounds related to acetaldehyde (vitisin B and ethyl linked A-T) can be easity detected by HPLC, did you measure them?

Line 338-345: Again the same observation, why authors did not present the quantitative results of the formation of Vitisin B and anthocyanins-ethyl-tannins? These values could help to demonstrate the formation of new colored compounds due to acetaldehyde and  tannins. If possible, these data should be presented.

Line 367: I can not understand how the addition of tannins to wine samples did not increase total phenol content or flavans reactive to vanillin. Please, explain

Figure 5: the statistical results among the different wine samples at t=0 and after Sat=4 should be presented to determine if there are significant differences in astringency according to the type of tannin.

Author Response

Abstract: The sentence “the astringency, determined before and at the end of the experiment, decreased in all the samples and mostly in the ET and GT samples” should be changed by “the astringency, determined before and at the end of the experiment, decreased in all the samples, included the control wine, and mostly in the ET and GT samples”

The amendment was made (line 31-32)

Line 39: some mention to Tea tannins should be done

The amendment was made (line 47-50)

Line 57-62: A comment regarding the practical aspect of this study could be really appreciated.

We thank the reviewer for this interesting suggestion, we added information on practical aspect about this work (line 88-90)

Line 68: malic acid

The amendment was made (line 97)

Line 120: Why Vitisin A, usually present at higher concentration than Vitisin B, was not quantified?

Line 330-336 The formation of compounds related to acetaldehyde (vitisin B and ethyl linked A-T) can be easity detected by HPLC, did you measure them?

Line 338-345: Again the same observation, why authors did not present the quantitative results of the formation of Vitisin B and anthocyanins-ethyl-tannins? These values could help to demonstrate the formation of new colored compounds due to acetaldehyde and  tannins. If possible, these data should be presented.

We thank the reviewer for these interesting suggestions but, we are sorry to say that the specifity of the method used which allowed the analysis of nine anthocyanins in 45 minutes, is not high for molecules in low amount as  Vitisin B and anthocyanins-ethyl-tannins as you could see in the chromatogram in the following figure related to the obtained chromatogram.

Therefore the ability to discriminate between vitisin A, vitisin B and all other anthocyanins derivatives (as the ethyl linked A-T) with a closely related structure eluting in the same narrow part of the chromatogram was poor. On the other hand, we couldn’t make a comparison with reference materials to be sure of the retention time of the specific anthocyanin derivative. We should have used another method with longer running times. However we thank the reviewer and we could consider it before planning the methods to be used in further experiments.

Line 367: I can not understand how the addition of tannins to wine samples did not increase total phenol content or flavans reactive to vanillin. Please, explain

In my opinion this is due to the fact that the amount of tannins added is very low compared to the great amount of total phenolic compounds and vanillin reactive flavans in an Aglianico wine. In addition, the sensitivity of the spectrophotometric methods used to determine phenolic compounds and vanilline reactive flavans is not so high to detect so little differences because in both cases a whole class of compounds were determined by exploiting only a common chemical reactivity.

Figure 5: the statistical results among the different wine samples at t=0 and after Sat=4 should be presented to determine if there are significant differences in astringency according to the type of tannin.

We added statistical difference in Figure 5

Reviewer 2 Report

The presented manuscript reports on the effect of structurally defined commercially available tannins from plant resources on different aspects of oxidation processes in red wine. The introduction gives a very short background and is backed up with a few relevant citations giving an overall compact entry point to the subject investigated. This part could be enriched by following considerations: The structural differences targeted in the study for the tannins used should be explained shortly to warranty the intended objectives. The influence of the specific wine phenolic composition should be elucidated in more detail e.g. which role would the other phenolic compounds may play beside the anthocyanins themselves. Please also explain why the production of major oxidation product acetaldehyde is important and how it may influence the wine quality. Finally, shortly elucidate the role of oxygen consumption in context of the sulfur dioxide turnover. These aspects are discussed later, but to reinforce the intended objectives, it would be more appropriate to have a general idea while reading the introduction. The methods are technically and scientifically sound to allow a good reproducibility of the results. The results are compactly presented giving the main relevant observations, with certain parts well reinforced in the discussion with the necessary corresponding citations. The submitted supplementary information should have a content list defining the extra material presented. Finally, since supplementary data is submitted, it should also be referred to in the manuscript (currently only one reference for tab. 2,3,4 is made, that is also not clear). Altogether the present work is a solid sum-up of some of the relevant data necessary for its technological application making it an interesting contribution.

Specific remarks:

Line 56: … acetaldehyde,. Please remove the “,”

Line 78: … by gently shaking the 4.5 L of wine in …

Line 98: … 35°C…

Line 149: In each session two tasting evaluations of four unknown samples were performed. Thereafter, there is a repetition of the sentence which should be corrected.

Line 167: … spray voltage 4.5 kV; please check other numbered digits and replace the comma with a point.

Line 203: … occurring in high enough amounts …

Line 207: … the occurrence of a number of molecule species in the mixture derived from the polymerization …

Line 212: … identified molecules to be solely constituted by EGCG and EGC …

Line 218: … in determining the number of galloyl subunits bound to…

Figure 2: Please mark the figures with A, B and C for each sub-diagram. It would be easier for the reader if the saturation times were given e.g. Sat-1 (12d) etc.

Line 273-275:  The sentence is difficult to follow – please rewrite with shorter sentences.

Line 275: … over the last saturation period.

Line 299: … value has already been detected during bottle aging…

Line 335: … malvidin and flavanols could be favoured…

Line 393: … does not have not a positive effect in reducing the …

Line 399: … tab 2,3,4,… figures?

Figure 5: Legend - … a statistically significant difference …

Reference should be properly formatted as required by the Journal preferences.

Author Response

The presented manuscript reports on the effect of structurally defined commercially available tannins from plant resources on different aspects of oxidation processes in red wine. The introduction gives a very short background and is backed up with a few relevant citations giving an overall compact entry point to the subject investigated. This part could be enriched by following considerations: The structural differences targeted in the study for the tannins used should be explained shortly to warranty the intended objectives. The influence of the specific wine phenolic composition should be elucidated in more detail e.g. which role would the other phenolic compounds may play beside the anthocyanins themselves. Please also explain why the production of major oxidation product acetaldehyde is important and how it may influence the wine quality. Finally, shortly elucidate the role of oxygen consumption in context of the sulfur dioxide turnover. These aspects are discussed later, but to reinforce the intended objectives, it would be more appropriate to have a general idea while reading the introduction.

All these information were added in the introduction. (Lines 56-71 new manuscript).

The methods are technically and scientifically sound to allow a good reproducibility of the results. The results are compactly presented giving the main relevant observations, with certain parts well reinforced in the discussion with the necessary corresponding citations. The submitted supplementary information should have a content list defining the extra material presented. Finally, since supplementary data is submitted, it should also be referred to in the manuscript (currently only one reference for tab. 2,3,4 is made, that is also not clear). Altogether the present work is a solid sum-up of some of the relevant data necessary for its technological application making it an interesting contribution.

The amendment was made through all the first part of Results and discussion and in the supplementary material added (see lines 46-50 and lines 233-294).

Line 56: … acetaldehyde,. Please remove the “,”

The amendment was made (line 81)

Line 78: … by gently shaking the 4.5 L of wine in …

The amendment was made (line 107)

Line 98: … 35°C…

The amendment was made (line 149)

Line 149: In each session two tasting evaluations of four unknown samples were performed. Thereafter, there is a repetition of the sentence which should be corrected.

Thanks to the reviewer suggestions, we removed the repetition sentence (line 201)

Line 167: … spray voltage 4.5 kV; please check other numbered digits and replace the comma with a point.

We checked all values and the sentence was changed as follows: spray voltage 4.5 kV, capillary temperature 350 °C, capillary voltage 30 V, sheath gas 20 and auxiliary gas 21 (line 220-222)

Line 203: … occurring in high enough amounts …

The amendment was made (line 258)

Line 207: ..the occurrence of a number of molecule species in the mixture derived from the polymerization..

The amendment was made (line 262)

Line 212: … identified molecules to be solely constituted by EGCG and EGC …

The amendment was made (line 267)

Line 218: … in determining the number of galloyl subunits bound to…

The amendment was made (line 273 

Figure 2: Please mark the figures with A, B and C for each sub-diagram. It would be easier for the reader if the saturation times were given e.g. Sat-1 (12d) etc.

We added this information in figures 2

Line 273-275:  The sentence is difficult to follow – please rewrite with shorter sentences.

The sentence was deleted.

Line 275: … over the last saturation period.

The amendment was made (line 331)

Line 299: … value has already been detected during bottle aging…

The amendment was made (line 355)

Line 335: … malvidin and flavanols could be favoured…

The amendment was made (line 394)

Line 393: … does not have not a positive effect in reducing the …

The amendment was made (line 453)

Line 399: … tab 2,3,4,… figures?

The amendment was made (line 459)

Figure 5: Legend - … a statistically significant difference …

The amendment was made in figure 5

Reference should be properly formatted as required by the Journal preferences.

The references have been modified and checked Inizio modulo

Reviewer 3 Report

Dear Authors,

The subject of study is of interest. I make the following comments, with the intention of improving the content.

Abstract

I believe that abstract could be improved, following the style of structured abstracts: 1) Background, 2) Methods, 3) Results and 4) Conclusion. Check the "instructions for authors" of the Molecules Journal.

Introduction

I miss a little bit more information about tannins used generally in wine making proccess. There are a few reviews, in the above issue, suitable to add in the introduction section. Check the literature avalaible.

Moreover, I find suitable to add a paragraph about the characteristics of Aglianico wine (type of grape, area of production, characteristics of the obtained wine ...). Furthermore, authors need to discuss in R & D section the obtained results with previous research carried out about Aglianico wine.

*NMR experiments are carried out in wine samples or in pure tannins used?

*HPLC and MS analyses. Calibration graphs, as well as Limit of Detection and Limit of quantification for each compound should be present in Results section.

*Abreviations used in Tables should be explained in the Table legend. Same for Figures.

*It could be great if the figures resolution/presentation is improved. They are with poor quality.

*Conclusions should be brief and in one paragraph.

*References need to be checked with the Journal style.

Author Response

 I believe that abstract could be improved, following the style of structured abstracts: 1) Background, 2) Methods, 3) Results and 4) Conclusion. Check the "instructions for authors" of the Molecules Journal.

The Abstract was changed as recommended. (line 12-32)

I miss a little bit more information about tannins used generally in wine making proccess. There are a few reviews, in the above issue, suitable to add in the introduction section. Check the literature avalaible.

The suggestion of the reviewer is more than right, and we added changed part of introductions as followed: 

Line 40-71.

The use of enological tannins is a common practice in winemaking, authorized by the International Organization of Vine and Wine (OIV) to facilitate the clarification of wines and musts. Also, they are used for many other purposes, such as inhibiting laccase in Botrytis-infected grapes [1], reducing the use of sulfur dioxide [2] and stabilizing the color of red wines [3].

In the wine industry, in addition to condensed tannins of grape origin, other commercial tannins are commonly used [4] including hydrolyzable tannins, gallotannins and ellagitannins, which are all extracted from different botanical species [5]. Ellagitannins are obtained from oak and chestnut wood [6], while gallotannins mainly from galla and tara nuts [7]. Tea tannins, well known for their beneficial bioactivities [8] were also tested in model wine solutions by Ugliano, Slaghenaufi, Picariello & Olivieri 2020 [9] who showed that they were capable of rapidly consume oxygen, although this is associated with the increased decline in the SO2 content.

Recently, some studies have reported on the importance of the use of enological tannins during oxidative processes. In model solutions was observed that: gallotannins have an excellent scavenging capacity while the ellagic improve iron(II) chelation and the condensed tannins have a significant ability to scavenge peroxyl radicals [10]. Moreover, the addition of ellagitannin in red winemaking operations increased the rate of O2 consumption [11]. The consumption of oxygen in wine is strictly related to phenolic compounds which are primary substrates for wine oxidation [12]. Wine oxidation occurs thanks to the presence in wine of transition metal ions and reducing species with a cathecol-like structure (o-diphenol groups). When oxidation reactions start the quinone is generated, and oxygen is reduced to hydrogen peroxide. Quinones and hydrogen peroxide are the main players of further processes of oxidation. In particular, hydrogen peroxide reacts with Fe2+ via the Fenton reaction to produce highly reactive hydroxyl radicals, which, in turn, oxidize additional wine constituents, including the most abundant, the ethanol to produce acetaldehyde, quinones react with several wine nucleophiles including thiols and the tannin phloroglucinol group [13Sulfur dioxide can react with quinones and acts as a quencher for the hydrogen peroxide stopping the oxidation process. On the other hand, when acetaldehyde is produced, several reactions with phenolics and other nucleophilic species occurred [14]. In red wines most of them involve anthocyanins and flavanols and can give ethyl linked phenolic compounds stabilizing red color and, often, changing the mouthfeel properties of wine tannins [15]. Obviously, differences in specific wine phenolic composition account for great differences among oxygen consumption rate and the effect on wine quality as observed when wines with different anthocyanins/tannins ratio underwent a controlled oxygen exposure. [16,17].

Moreover, I find suitable to add a paragraph about the characteristics of Aglianico wine (type of grape, area of production, characteristics of the obtained wine ...). Furthermore, authors need to discuss in R & D section the obtained results with previous research carried out about Aglianico wine.

We added a paragraph about the characteristics of Aglianico wine (line 117-131)

NMR experiments are carried out in wine samples or in pure tannins used?

The NMR experiments was carried out in pure tannins (line 235)

HPLC and MS analyses. Calibration graphs, as well as Limit of Detection and Limit of quantification for each compound should be present in Results section.

We added in the material and method paragraph this information:

For MS analysis:

Line (222-223)  The LOD was determined as a signal-to-noise ratio (S/N) of 3:1 (30 ng mL-1 for catechin, and  40 ng mL-1 for epicatechin).

For HPLC analysis:

Lines (170-173) The calibration curve was characterized by a correlation coefficient (r2) > 0.998. The LOD was determined as a signal-to-noise ratio (S/N) of 3:1 (0.0262 mg L-1), and the LOQ was determined as a S/N of 10:1 (0.0866 mg L-1).

Abbreviations used in Tables should be explained in the Table legend. Same for Figures.

Thanks to the reviewer suggestions, a new table was added Tab 1

It could be great if the figures resolution/presentation is improved. They are with poor quality.

The amendment was made

Conclusions should be brief and in one paragraph.

We thank the reviewer for this interesting suggestion, we changed the paragraph and we contracted the conclusion (line 499-510)

References need to be checked with the Journal style.

The references have been modified and checked Inizio modulo

Round 2

Reviewer 1 Report

After reading your comments and the changes made in the article, I think it can be now accepted for publication.

Reviewer 3 Report

Dear Authors,

The MS was improved. My questions and comments were addressed.